# Effect of Fiber Side-Feeding on Various Properties of Nickel-Coated Carbon-Fiber-Reinforced Polyamide 6 Composites Prepared by a Twin-Screw Extrusion Process

**Naeun Jeong and Donghwan Cho \***

Department of Polymer Science and Engineering, Kumoh National Institute of Technology, Gumi 39177, Gyeongbuk, Republic of Korea
* Correspondence: dcho@kumoh.ac.kr

**Abstract:** In the present study, how side-feeding of NiCF during twin-screw extrusion processing influences the fiber aspect ratio and thermal, mechanical, electrical, and electromagnetic properties of nickel-coated carbon fiber (NiCF)-reinforced polyamide 6 (PA6) composites was explored. For this, the fiber length distribution, thermal stability, heat deflection temperature, dynamic mechanical property, tensile, flexural, electrical resistivity, and electromagnetic interference shielding effectiveness (EMI SE) properties of NiCF/PA6 composites were extensively investigated. Chopped NiCF was regularly fed via either a main feeder or a side feeder and NiCF/PA6 pellets with different fiber-feeding pathways were prepared. The side-feeding effect of NiCF on the fiber length distribution and the composite properties was studied. The thermal stability, heat deflection temperature, storage modulus, tensile, flexural, and surface resistivity, and EMI SE properties of the NiCF/PA6 composites strongly depended not only on the NiCF content but also on the feeding method (main-feeding or side-feeding) upon extrusion processing, indicating that the fiber length distribution relevant to the fiber aspect ratio was critically important to enhance the composites' properties. As a result, the NiCF/PA6 composites produced via side-feeding of NiCF exhibited an NiCF distribution longer than that produced via main-feeding, leading to enhancement of the thermal stability, heat deflection temperature, storage modulus, tensile, flexural, and EMI SE properties, strongly depending on the NiCF content.

**Keywords:** nickel-coated carbon fiber; polyamide 6; composite; extrusion; side-feeding; properties

## 1. Introduction

Nowadays, electronic devices provide many conveniences and benefits in our daily life. On the other hand, electromagnetic interference (EMI), which can be derived from them, may cause operational malfunctions of electronic devices and negatively impact on human health [1,2]. Hence, research and development of materials, which can shield EMI, have progressed with concerns about and consciousness of EMI shielding issues in many advanced countries [3,4].

Conventionally, metal-based materials have been widely utilized for EMI shielding. They have been increasingly substituted with carbon-based materials. Carbon-based materials such as carbon fiber, graphite, carbon nanotube, graphene, etc., have often been used for academic and industrial purposes because they can provide EMI shielding effectiveness (EMI SE), depending on the material used [5–7]. Recently, many studies have been performed to develop fiber-reinforced composite materials for shielding or reducing EMI [5,8,9].

Carbon-fiber-reinforced composites, which have a low density and easy processability in comparison to metal-based materials, exhibit excellent mechanical, thermal, and electrical properties as well as EMI shielding performance [10–12]. Carbon fiber normally exhibits higher electrical resistivity than metals. Therefore, upon composite fabrication, a large

amount of carbon fiber should be used to have an EMI SE comparable to metals. Considering processing difficulties due to the increased amount of carbon fiber, chopped carbon fibers thinly coated or plated with metals such as nickel and copper can be frequently used to decrease electrical resistivity and to increase EMI SE [13–16]. Composites consisting of metal-coated carbon fiber and thermoplastic polymers can be produced by conventional extrusion and injection processes [11,17]. In extrusion processes, increased carbon fiber loading results in enhanced EMI SE, whereas the processability and the material toughness can be significantly decreased as well.

Nickel has often been used to coat carbon fibers because it is less expensive and less oxidative with good bonding capability to the polymer matrix [15]. In general, nickel coating to carbon fiber is conducted after primary copper coating is carried out on the fiber surface. Copper is highly conductive, but easily oxidized. Therefore, secondary nickel coating is performed to impart the conductivity to the copper-coated carbon fiber. Accordingly, it is referred to as nickel-coated carbon fiber (NiCF) because the copper layer is not shown from the outside [18]. Metal-coating of carbon fiber can often be performed by means of metal-plating, fiber-metal co-extrusion, etc. Metal-plating is a manufacturing process in which a thin layer of metal coats a substrate. It is usually carried out by immersing the metal in an acid solution with an anode electric current and cathode. The material to be plated is made from the cathode of an electrolysis cell through which a direct electric current is passed. The anode is usually the metal being plated.

It is known that NiCF exhibits not only good electrical and electromagnetic properties, but also good mechanical and thermal properties [18,19]. NiCF can be used to fabricate NiCF-reinforced composites with thermoplastic polymers such as polyethylene, polypropylene, polyamide, acrylonitrile–butadiene–styrene, etc., via various composite fabrication methods such as extrusion, injection, and compression molding processes [17,20–23].

In our previous work, it was found that the fiber-feeding efficiency can importantly influence the electrical properties as well as the mechanical and thermal properties of carbon fiber/polypropylene composites [11]. This gave us research motivation not only to extend this to other composites with engineering plastics, which can be produced by a twin-screw extruder with different feeding methods of NiCF, but also to improve the relevant properties of the resulting carbon-fiber-reinforced engineering plastic matrix composites.

Polyamides are representative engineering plastics, which can be used for engineering purposes due to their excellent mechanical, thermal, and wear-resistant performance and good processability [24]. Many industrial parts made with polyamides have been manufactured by extrusion and injection molding processes. Over the last decades, many reports have dealt with composites made of carbon fiber and polyamide [25–27]. Several studies on the processing and properties of composites consisting of metal-coated carbon fiber and a thermoplastic matrix have been performed [21,28,29]. However, the side-feeding effect of carbon fiber during the twin-screw extrusion process on various properties of NiCF-reinforced polyamide matrix composites has rarely been studied.

Consequently, the objective of the present study is to diagnose how side-feeding of NiCF during twin-screw extrusion processing influences the fiber aspect ratio and the thermal, mechanical, electrical, and electromagnetic properties of the resulting NiCF/polyamide 6 composites. For this, the fiber length distribution, thermal stability, heat deflection temperature, dynamic mechanical property, tensile, flexural, electrical resistivity, and EMI SE properties of NiCF/polyamide 6 composites were extensively investigated. In addition, the results obtained from the main-feeding and side-feeding processes were compared and discussed.

## 2. Materials and Methods

### 2.1. Materials

A polyacrylonitrile-based carbon fiber (12K, T700 Grade), which was commercially manufactured by Toray Advanced Materials Co., Gumi, Republic of Korea, was used in this work. The average diameter of the single carbon fiber filament was about 7 μm. The

density was about 1.8 g/cm$^3$. Chopped NiCF of 6 mm in length, which was supplied by BSM Advanced Materials Co., Jeongeup, Republic of Korea, was used as reinforcement for the NiCF/polyamide 6 composites throughout this work. Nickel coating of the carbon fiber was performed by using a nickel-plating method at BSM Advanced Materials Co., Jeongeup, Republic of Korea. The average thickness of the coated nickel was in the range of 250–300 nm. The density of the NiCF was 2.8 g/cm$^3$. Polyamide 6 (PA6) pellets (Model KN136), which were supplied by Kolon Plastics Co., Gimcheon, Republic of Korea, were used as the matrix of the composite. According to the manufacturer's information, the melt flow index of the 'as-received' PA6 was 34.2 g/10 min, the glass transition temperature was 44 °C, the specific gravity was 1.14, the number average molecular weight was 40,000, and the melting temperature was 222 °C. The supplied NiCF and PA6 pellets were fully dried in an air-circulating convection oven prior to use.

### 2.2. Processing of the NiCF/PA6 Pellets and the Composites

The NiCF/PA6 pellets and the composites were produced by twin-screw extrusion and injection molding processes, respectively, by controlling the processing parameters such as the barrel temperatures, screw speed, and feeding rate. The optimal processing parameters were found through preliminary processing work.

Two different kinds of extruders were used for the extrusion process in the present work. For main-feeding (MF) of both chopped NiCF and PA6 pellets, a twin-screw extruder of a modular intermeshing co-rotating type (BA-11, Bautek Co., Pocheon, Republic of Korea) with a L/D ratio of 40 and each screw with a diameter of 11 mm was used, as depicted in Figure 1A. For side-feeding (SF) of the chopped NiCF and by main-feeding of the PP pellets, a co-rotating twin-screw extruder (Model PCM30, LG Machinery Co., Seoul, Republic of Korea) with an L/D ratio of 38 and each screw with a diameter of 30 mm, and the equipped side feeder (WCA-302) was used, as illustrated in Figure 1B.

In both feeding cases, the extrudates were continuously cooled down in a water bath and then cut to 2–3 mm using a pelletizer. The processing temperature was varied in the range of 250–290 °C according to the screws' location in the barrel. NiCF/PA6 pellets of 5 kg, which were sufficient to make the composites for the present work, were obtained via MF and SF, respectively. The NiCF/PA6 ratios of the pellets prepared from each twin-screw extruder were 0/100, 10/90, 20/80, and 30/70 by weight.

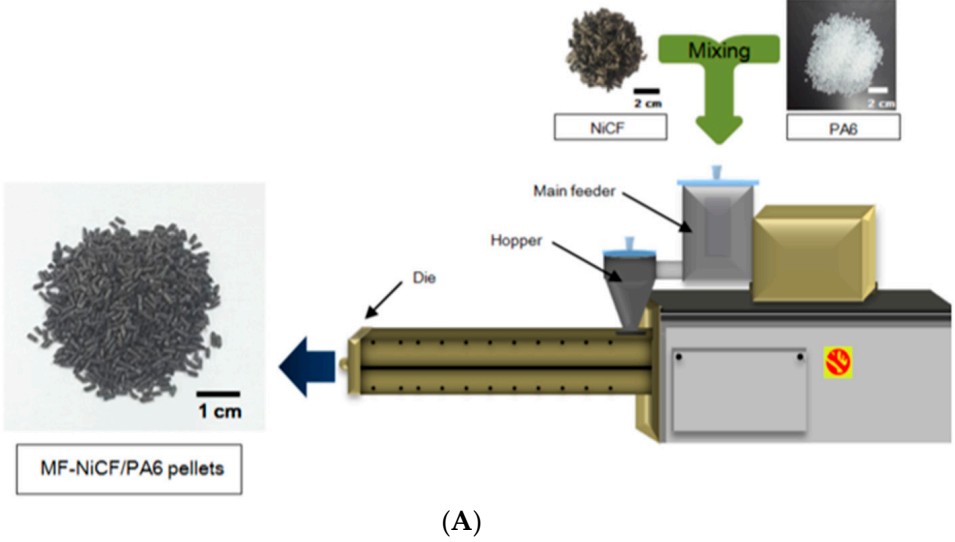

**(A)**

**Figure 1.** *Cont.*

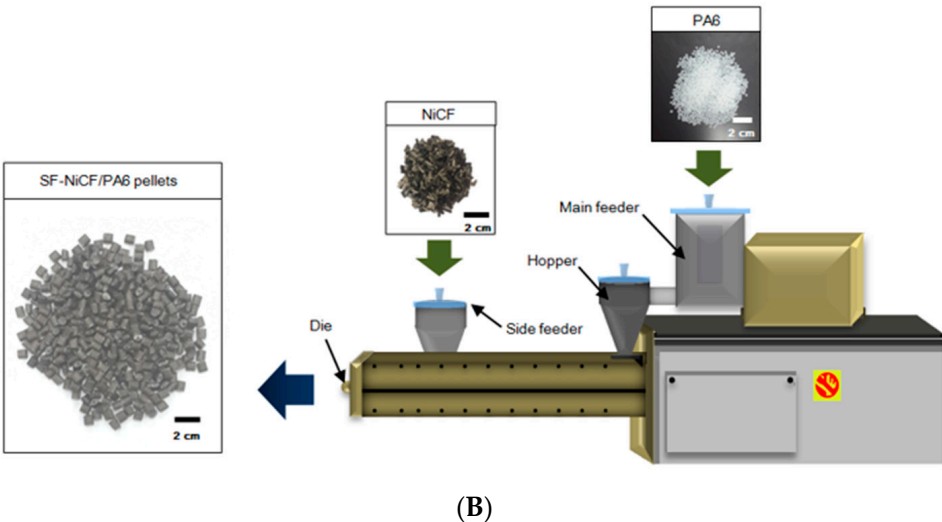

(**B**)

**Figure 1.** Extrusion processing to produce (**A**) MF-NiCF/PA6 and (**B**) SF-NiCF/PA6 pellets using a twin-screw extruder.

After sufficiently drying the pellets, NiCF/PA6 composites with various NiCF contents were produced by an injection molding process (PRO-WD 80, Dong Shin Co., Changwon, Republic of Korea) using the pellets prepared via MF and SF, respectively. Hereinafter, the composites, which were made with the pellets prepared via MF and SF of NiCF upon extrusion, were designated as "MF-NiCF/PA6" and "SF-NiCF/PA6", respectively. The injection-molded composites were used as specimens for the heat deflection temperature, dynamic mechanical, tensile, and flexural tests.

### 2.3. Preparation of NiCF/PA6 Composites for EMI SE Measurement

The EMI SE specimens were prepared by compression molding (GE-122S, Kukje Scien, Co., Daejeon, Republic of Korea) using the NiCF/PA6 pellets dried at 80 °C for 6 h, as shown in Figure 2. According to the ASTM D4935-10 standard [30], the pellets of about 15 g were placed uniformly in a stainless-steel mold, heated up to 240 °C for 8 min, and then pressed up to 6.9 MPa (1000 psi). During compression molding, a couple of debulking steps were carried out to remove pores and voids possibly existing in the composite. The reference and load specimens with a thickness of about 0.7 mm for EMI SE measurement, as indicated in Figure 2, were obtained by varying the NiCF content.

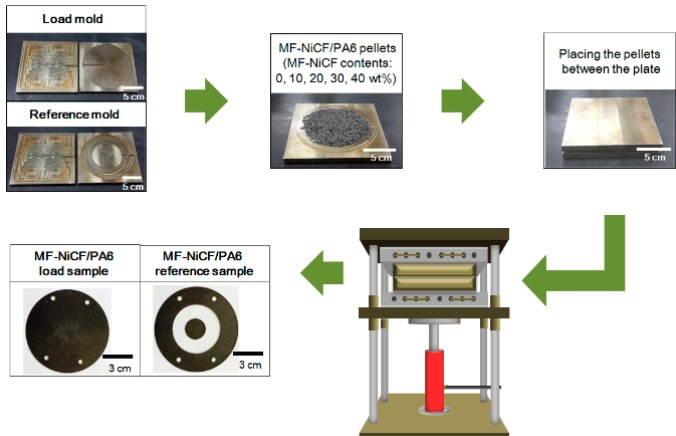

**Figure 2.** Compression molding process to prepare the load specimen and the reference specimen with various NiCF contents for EMI SE measurement.

### 2.4. Fiber Length Distribution Measurement

The fiber length distribution of the NiCF/PA6 composites was investigated using a scanning electron microscope (JSM 6380, JEOL, Tokyo, Japan) to measure the length of NiCF in each composite. The individual fibers were obtained by dissolving the composite sample in a solvent and then by separating the fibers from the PA6 matrix part. About 200 fibers were used to examine the fiber length distribution of each composite. To observe the fiber length by SEM, NiCF was coated with platinum by a sputtering method.

### 2.5. Thermal Characterization

The thermal stability of neat PA6 and NiCF/PA6 pellets with various fiber contents prepared by different fiber-feeding methods upon extrusion was measured up to 800 °C under a nitrogen atmosphere by means of thermogravimetric analysis (TGA Q500, TA Instruments, New Castle, DE, USA). The heating rate was 20 °C/min. About 12 mg of the sample was loaded into an alumina can for each analysis.

The heat deflection temperature (HDT) of the neat PA6 and NiCF/PA6 composites with various fiber contents prepared by different fiber-feeding methods was measured using an HDT tester (Tinius Olsen, Model 603, Horsham, PA, USA) according to the ASTM D648M standard [31]. The test was performed in a silicone oil chamber using a three-point bending mode, applying a load of 0.455 MPa. The silicone oil in the chamber was heated with a heating rate of 120 °C/h after the specimen was immersed in the chamber for 3 to 5 min. The specimen dimensions were 125 mm × 12.5 mm × 3 mm. The HDT was recorded at the temperature where the sample was deflected by 0.254 mm during the measurement. The average HDT of each composite was obtained from three specimens.

The storage modulus of the neat PA6 and NiCF/PA6 composites with various fiber contents prepared by different fiber-feeding methods was examined with the heating rate of 3 °C/min in air by means of dynamic mechanical analysis (DMA Q800, TA Instruments, New Castle, DE, USA). A dual-cantilever mode with a drive-clamp and a fixed-clamp was used. The measuring temperature range was from 25 to 200 °C. The oscillation amplitude was 10 μm. The frequency was 1 Hz. The dimensions of the rectangular-shaped specimens were 63.5 mm × 12.5 mm × 3 mm.

### 2.6. Mechanical Tests

Tensile tests were performed with neat PA6 and NiCF/PA6 composites with various fiber contents prepared by different fiber-feeding methods using a universal testing machine (UTM, AG-50kNX, Shimadzu JP Co., Kyoto, Japan) according to the ASTM D638M standard [32]. The crosshead speed was 50 cm/min and the load cell of 50 kN was used. Dog-bone-shaped specimens with the dimensions of 150 mm × 12.5 mm × 3 mm were used. The average tensile modulus and strength were obtained from 10 specimens per composite.

Three-point flexural tests were also performed with neat PA6 and NiCF/PA6 composites with various fiber contents prepared by different fiber-feeding methods using a universal testing machine (UTM, AG-50kNX, Shimadzu JP Co., Kyoto, Japan) according to the ASTM D790M standard [33]. The span-to-depth ratio was 32. The crosshead speed was 5.1 mm/min. The rectangular-shaped specimens had the dimensions of 12.5 mm × 12.5 mm × 3 mm. The average flexural modulus and strength were obtained from 10 specimens per composite.

### 2.7. Surface Resistivity and EMI SE Measurements

The surface resistivity of the neat PA6 and NiCF/PA6 composites with various fiber contents prepared by different fiber-feeding methods was measured using a surface resistivity tester with a two-point probe (ACL 800 Megohmmeter, ACL Staticide Inc., Chicago, IL, USA). Three specimens per composite were used for each test. The EMI SE of each composite specimen was also measured at ambient temperature using network analysis (E5071C, Agilent, Santa Clara, CA, USA) and a standard test fixture of interference shielding effectiveness (EM-2017A, ELECTRO-METRICS, Johnstown, NY, USA). The frequency range

used was 30 MHz~1.5 GHz. The EMI SE was obtained by comparing the power of the reference sample and that of the load sample according to the ASTM D4935-10 standard [30]. The reference sample was to measure the intensity in the absence of EMI shielding material, whereas the load sample was to measure the intensity in the presence of EMI shielding material. The procedure was repeated by varying the load sample. The differences in the signal occurring between the reference sample and the load sample were recorded.

## 3. Results and Discussion

### 3.1. Fiber Length Distribution

Figure 3 displays the length distribution histogram of NiCF existing in the MF-NiCF/PA6 and SF-NiCF/PA6 composites. In the MF-NiCF/PA6 composites, the NiCF ranging 150–200 μm exhibited the highest fiber counts with about 30% of the total, indicating the average length of 191 μm. The shortest fiber length was about 62 μm, and the longest fiber length was about 350 μm. NiCF longer than 350 μm were not found. In the SF-NiCF/PA6 composite, the NiCF in the range of 200–250 μm exhibited the highest fiber counts with about 35%, indicating the average length of 233 μm. The shortest fiber length was about 141 μm, and the longest fiber length was about 444 μm. NiCF shorter than 140 μm were not found.

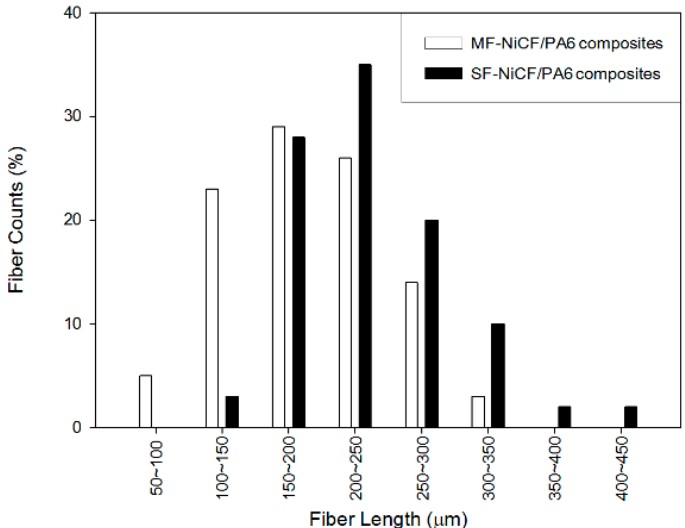

**Figure 3.** A histogram showing the fiber length distribution of the MF-NiCF/PA6 and SF-NiCF/PA6 composites.

The results indicate that the aspect ratio of NiCF in the composites after extrusion/injection processing was much higher in the SF-NiCF/PA6 composites than in the MF-NiCF/PA6 composites. In other words, the average length of carbon fibers in the composite produced via side-feeding was longer than that in the composite produced via main-feeding. This can be explained by the following. The fiber damage and length degradation can result from the mechanical shear forces by a screw-motion in the barrel during the extrusion process. Side-feeding of the chopped carbon fibers resulted in the fiber path length being shorter than the main-feeding length, with it experiencing less mechanical shear by the twin screws and a longer residence time in the barrel during extrusion. Accordingly, it may be expected that the fiber length distribution may importantly influence not only the thermal and mechanical properties but also the surface resistivity and EMI SE of the resulting composites.

Figure 4 shows typical SEM images showing the fiber length variation observed from the MF-NiCF/PA6 and SF-NiCF/PA6 composites, respectively. As mentioned above, the individual fibers were obtained by dissolving the composite sample in a solvent and then by separating the fibers from the PA6 matrix part. As seen above, the average length of the

NiCF in the MF-NiCF/PA6 composite was longer than that in the SF-NiCF/PA6 composites. The SEM images qualitatively support the fiber length distribution described in Figure 3.

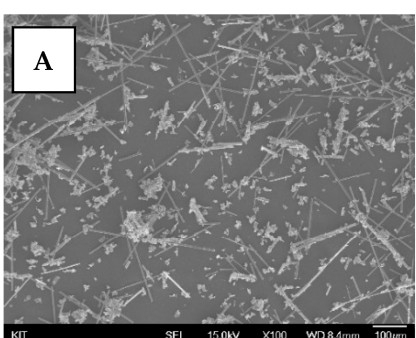 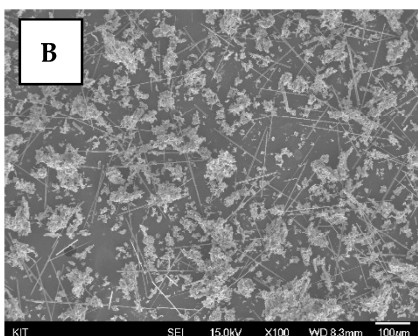

**Figure 4.** SEM images showing the length variation of NiCF in the (**A**) MF-NiCF/PA6 and (**B**) SF-NiCF/PA6 composites.

### 3.2. Thermal Stability

Figure 5 shows the thermal stability of the neat PA6 and NiCF/PA6 pellets. With closer inspection of the inserted figure, it was found that the thermal stability in the range of 350–450 °C was slightly affected by the fiber contents and different fiber-feeding methods. In the figures, MF-10, MF-20, and MF-30 designate the composites prepared by main-feeding of 10, 20, and 30 wt% NiCF, respectively, whereas SF-10, SF-20, and SF-30 designate the composites prepared by side-feeding of 10, 20, and 30 wt% NiCF, respectively. The TGA curves indicate that the residual weights at 600 °C measured with each composite were approximately agreed with the weight percent of NiCF contained in the pellets, that is, 0, 10, 20, and 30 wt%. The result shows that the thermal stability was increased in the order of MF-0 (neat PA6), MF-10, MF-20, SF-10, SF-20, MF-30, and SF-30. The SF-NiCF/PA6 pellets containing fibers longer than the MF-NiCF/PA6 pellets exhibited increased thermal stability when the NiCF contents were corresponding to each other. It seems that both the fiber content and the fiber length distribution were responsible for the increased thermal stability.

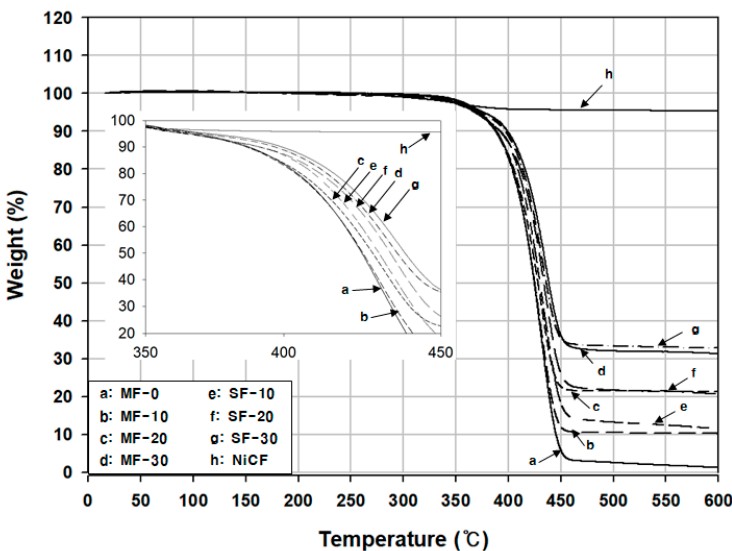

**Figure 5.** TGA curves of the MF-NiCF/PA6 and SF-NiCF/PA6 pellets with various NiCF contents.

### 3.3. Heat Deflection Temperature

Figure 6 represents the effect of NiCF content and the fiber-feeding method on the HDT of the MF-NiCF/PA6 and SF-NiCF/PA6 composites. The HDT of neat PA6 was about 159 °C. The HDT was highly increased to 186 °C with 10 wt% MF-NiCF and 194 °C with

10 wt% SF-NiCF. The HDT was gradually increased up to 208 °C by increasing the NiCF content resulting in an increase of 26 to 49 °C, compared to that of the neat PA6, depending on the NiCF content. This was ascribed to the fiber-reinforcing effect resisting the three-point bending load applied during the measurement. The SF-NiCF/PA6 composites had higher HDTs than their MF-NiCF/PA6 counterparts because the reinforcing effect was more pronounced in the SF case with the high fiber aspect ratio than in the MF case with the relatively short fiber aspect ratio. Reinforcing fibers with a higher aspect ratio can more efficiently transfer the applied load to neighboring fibers through the matrix [34]. Cheremisinoff et al. [35] reported that the HDT was linearly increased by increasing the reinforcing fiber up to 20 wt%, and such the increase was similarly found with milled fiber reinforcement.

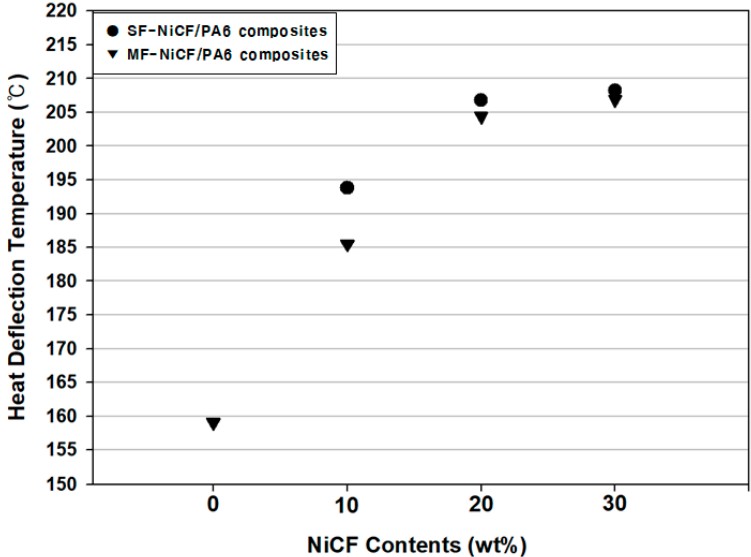

**Figure 6.** HDT values of the MF-NiCF/PA6 and SF-NiCF/PA6 composites with various NiCF contents.

*3.4. Storage Modulus*

Figure 7 shows the variation in the storage modulus measured with the neat PA6 and NiCF/PA6 composites, depending on the NiCF content and the fiber-feeding method. The storage modulus rapidly decreased with the increasing temperature, particularly above about 40 °C due to the glass transition behavior of neat PA6 and the PA6 matrix of the composite. The storage modulus of the neat PA6 at 25 °C was about 1.9 GPa. It was gradually increased by increasing the NiCF content, indicating the highest modulus of 6.1 GPa with the 30 wt% SF-NiCF/PA6 composite and the highest modulus of 5.1 GPa with the 30 wt% MF-NiCF/PA6 composite, indicating a 19% improvement by side-feeding of NiCF.

The increased improvement in the SF-NiCF/PA6 composite can be explained by the increased reinforcing effect due to the length distribution and the aspect ratio of NiCF in the composite, as described earlier. It may be said that the PA6 matrix surrounding the individual fibers of the composite that was able to distribute the external load to the neighboring fibers and the reinforcing fibers with a high aspect ratio could withstand the frequency applied under the dual-cantilever mode during dynamic mechanical analysis. The increased NiCF content was also responsible for the increased dynamic mechanical property. Rezaeis et al. [36] found that the storage modulus of a carbon fiber/polypropylene composite was increased by increasing the carbon fiber length from 0.5 to 10 mm.

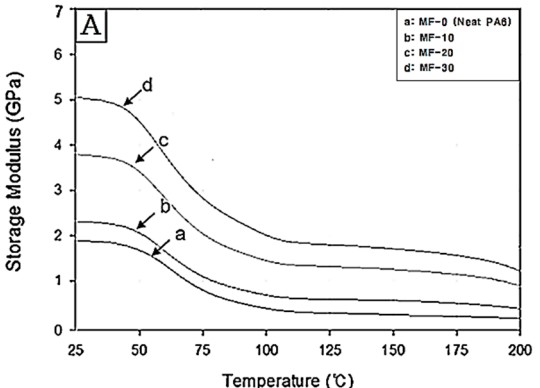
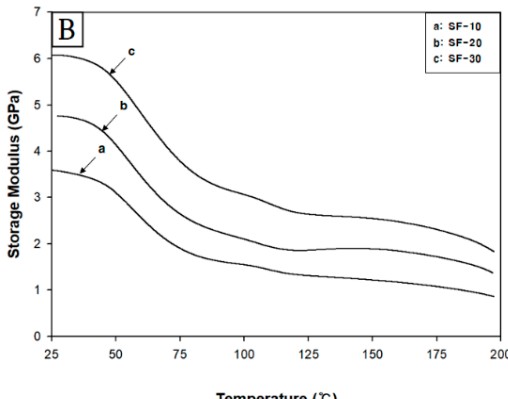

**Figure 7.** Variation in the storage modulus of the (**A**) MF-NiCF/PA6 and (**B**) SF-NiCF/PA6 composites with various NiCF contents.

### 3.5. Tensile Properties

Figure 8 compares the tensile modulus and strength of the NiCF/PA6 composites between MF and SF, depending on the NiCF content. The neat PA6 (MF-0) exhibited a tensile modulus of 2.2 GPa and thea tensile strength of 76 MPa. By increasing the NiCF content, the tensile modulus and strength of the NiCF/PA6 composites were markedly increased due to the reinforcing effect of NiCF. The tensile modulus and strength were about 6.9–7.3 GPa and 175–184 MPa, respectively. The values exhibited the highest values, with them being improved by about 210% and 140%, respectively, when the neat PA6 was reinforced with 30 wt% NiCF and the composite was produced via side-feeding of NiCF. With the same fiber content, the tensile modulus of the SF-NiCF/PA6 composites was 5–20% higher than that of the MF-NiCF/PA6 composites according to the NiCF content, as shown in Figure 8A. Meanwhile, the tensile strength of the SF-NiCF/PA6 composites was 15–50% higher than that of their MF-NiCF/PA6 counterparts depending on the NiCF content, as indicated in Figure 8B. The percent improvement of the tensile strength by side-feeding of NiCF was higher than that by main-feeding. This can be explained by the fact that the average length of the carbon fibers existing in the composite produced via side-feeding was longer than that in the composite produced via main-feeding, as indicated in Figure 3.

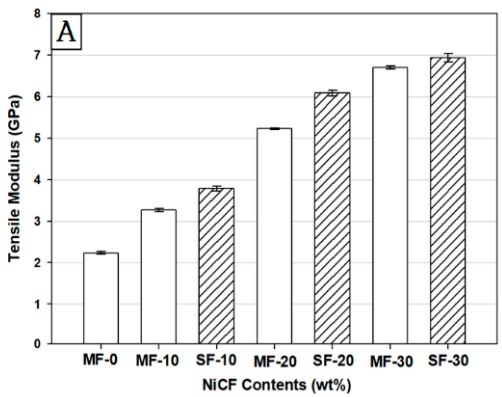
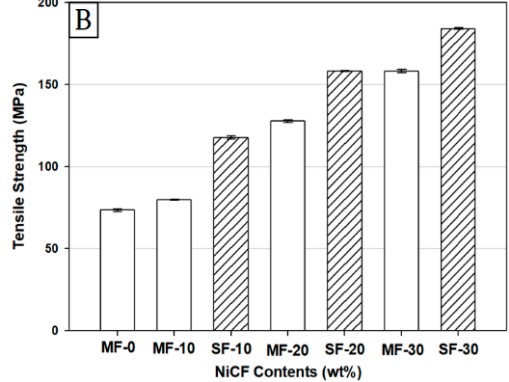

**Figure 8.** Comparisons of (**A**) the tensile modulus and (**B**) the tensile strength of the MF-NiCF/PA6 and SF-NiCF/PA6 composites with various NiCF contents.

### 3.6. Flexural Properties

Figure 9 shows the flexural modulus and strength of the NiCF/PA6 composites produced via MF and SF, respectively. Both the flexural modulus and strength strongly depended not only on the carbon fiber content but also on the feeding method. The neat PA6 exhibited a flexural modulus of 1.7 GPa and a flexural strength of 49 MPa. The flexural

properties of the composites were gradually increased by increasing the NiCF, as similarly found in the tensile results. By increasing the NiCF content, the flexural modulus and strength were remarkably increased up to 11.4 GPa and 197 MPa in the case of SF-30 and 10.4 GPa and 165 MPa in the case of MF-30, respectively. It was obvious that the SF-NiCF/PA6 composite exhibited a flexural modulus and strength higher than its MF-NiCF/PA6 counterpart. In the case of side-feeding of 30 wt% NiCF, the flexural modulus was highly improved by about 570% and the strength increased by about 240%, compared to that of the neat PA6.

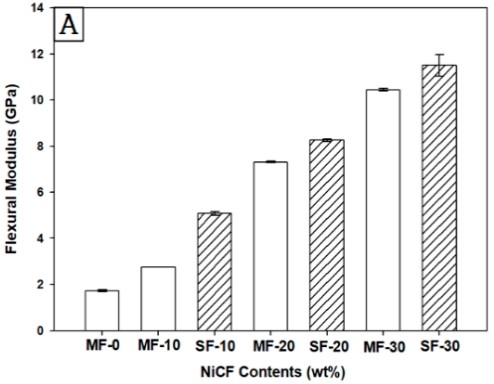
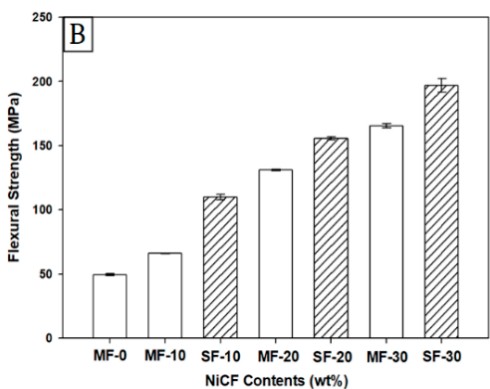

**Figure 9.** Comparisons of (**A**) the flexural modulus and (**B**) the flexural strength of the MF-NiCF/PA6 and SF-NiCF/PA6 composites with various NiCF contents.

Figure 10 displays representative load versus displacement curves measured with MF-NiCF/PA6 and SF-NiCF/PA6 composites with various NiCF contents. As shown, each curve was consistent with the variation in the flexural modulus and strength of the MF-NiCF/PA6 and SF-NiCF/PA6 composites with various NiCF contents, as shown in Figure 9.

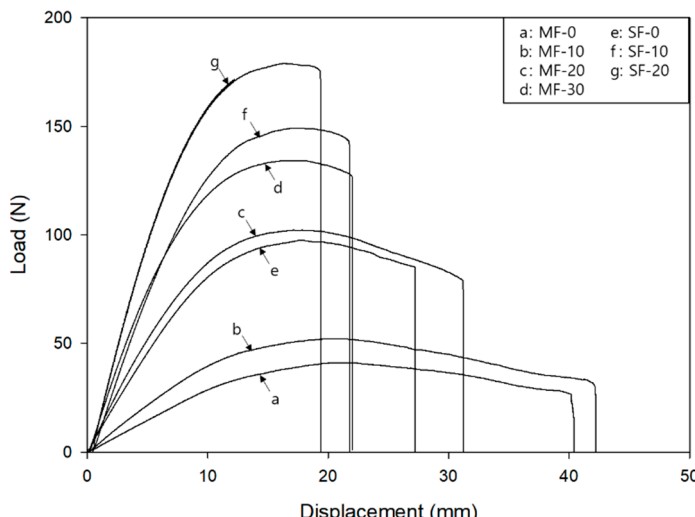

**Figure 10.** Load versus displacement curves measured with the MF-NiCF/PA6 (a–d) and SF-NiCF/PA6 (e–g) composites with various NiCF contents.

As found in Figures 8 and 9, the tensile and flexural properties of the NiCF/PA6 composites were remarkably increased by increasing the NiCF content when side-feeding of the chopped carbon fibers was adopted. This may be explained by the metal-coated carbon fibers, shearing force, and fiber moving distance during compounding upon extrusion. Metal coating of the carbon fiber made the fiber filament stiffer. Chopped NiCF could be easily exposed to the shearing force occurring between the barrel and the screw during

compounding, as similarly found in other composite systems [37,38]. In the case of main-feeding, chopped NiCF with the melted PA6 resin could move from the main-hopper entrance to the die-end of the extruder. Meanwhile, chopped NiCF could move from the side-feeder entrance to the die-end with a shorter fiber moving path in the side-feeding case than in the main-feeding case. Shortening and damaging of brittle NiCF occurred to some extent during the extrusion process. Such shortening of and damage to the fiber was found in the main-feeding method more than in the side-feeding method, as described earlier.

Such phenomena were directly responsible for the lowering of the fiber aspect ratio in the resulting composite. The decreased tensile modulus and strength were attributed to the lowered fiber aspect ratio. The carbon fiber length distribution results shown in Figure 3 well support both the tensile and flexural results. The fiber length distribution and the aspect ratio significantly influence the mechanical properties of a composite material. When the fiber length distribution and the fiber aspect ratio were increased, the composite could resist external forces more effectively by transferring the external forces to neighboring fibers surrounded by the matrix and by distributing the applied mechanical load to the individual reinforcing fibers. As a result, the mechanical properties of the composite with a long fiber distribution and a high aspect ratio were enhanced, compared to those with a short fiber distribution and a low aspect ratio with the same fiber contents. Several papers [38–41] have reported that the length distribution and the aspect ratio of reinforcing fibers significantly influence the tensile and flexural properties of fiber-reinforced polymer matrix composites produced via an extrusion process.

### 3.7. Surface Resistivity

Figure 11 depicts the variation in the surface resistivity of the MF-NiCF/PA6 and SF-NiCF/PA6 composites as a function of NiCF contents. In general, the surface resistivity of non-conductive polymers is decreased in the presence of conductive fillers, depending on the filler concentration therein [42]. The surface resistivity of the neat PA6 was not able to be measured because it was out of the measuring range of the resistivity tester with a two-point probe used in this work. As shown, the surface resistivity of the NiCF/PA6 composites was considerably decreased by increasing the NiCF content. The SF-NiCF/PA6 composite exhibited a surface resistivity lower than its MF-NiCF/PA6 counterpart. Although the difference in the resistivity with 10 wt% NiCF between the two composites prepared via different fiber-feeding methods was not highly distinguishable, the distinction was profound with 20 and 30 wt% NiCF. Meaningfully, the result stressed that the appropriate incorporation of NiCF into the PA6 matrix by side-feeding of NiCF gave rise to the electrical conductivity of the resulting composite.

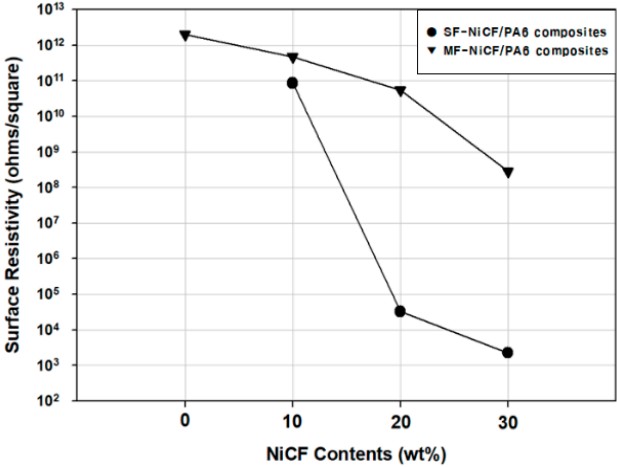

**Figure 11.** Surface resistivity of the MF-NiCF/PA6 and SF-NiCF/PA6 composites with various NiCF contents.

It was believed that the increased fiber aspect ratio due to side-feeding significantly contributed to lowering the percolation threshold, resulting in an increase in the electrical pathway in the NiCF/PA6 composite. The decreased surface resistivity by side-feeding of NiCF may play a positive role in the EMI SE of the NiCF/PA6 composite as well.

In several earlier papers [17,43,44], it was reported that the fiber aspect ratio can significantly influence not only the surface resistivity but also the EMI SE of carbon-fiber-reinforced polymer composites. The electrical conductivity and the EMI SE of carbon=fiber-reinforced polymer composites strongly depend on the type, content, geometrical shape, and dispersion of conductive fillers such as carbon fibers, carbon nanotubes, and graphene, altering the electrical conduction mechanism.

### 3.8. Electromagnetic Interference Shielding Effectiveness

Figure 12 displays the effect of NiCF side-feeding on the EMI SE of the MF-NiCF/PA6 and SF-NiCF/PA6 composites with different NiCF contents as a function of frequency. The neat PA6 exhibited nearly 0 dB in all of the frequency ranges, as expected. Meanwhile, the EMI SE of the NiCF/PA6 composites was influenced by the fiber-feeding method as well as the fiber content.

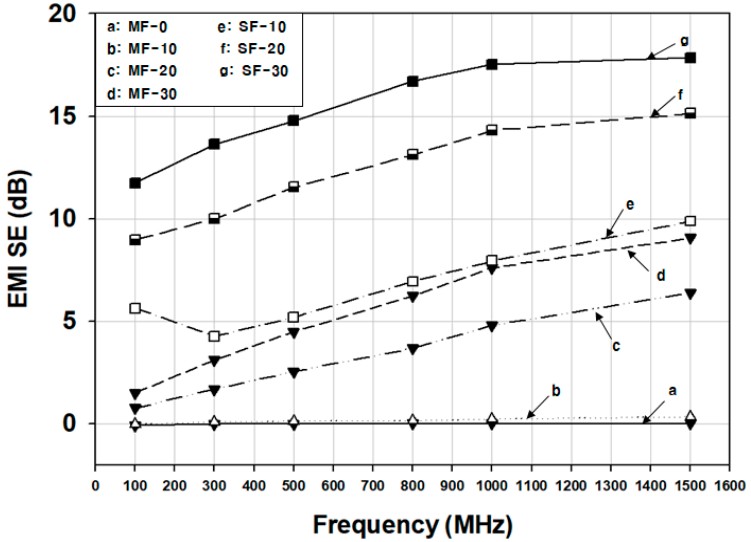

**Figure 12.** EMI SE variation in the MF-NiCF/PA6 and SF-NiCF/PA6 composites with various NiCF contents as a function of frequency.

The EMI SE of the MF-NiCF/PA6 composite was gradually increased by increasing NiCF and with the frequency, with it expected at 10 wt%. The EMI SE value at 1500 MHz was markedly increased by 29 times from 0.3 dB with 10 wt% NiCF to 9 dB with 30 wt% NiCF. The EMI SE of the SF-NiCF/PA6 composite was further increased compared to that of the MF-NiCF/PA6 composite. The value at 1500 MHz was increased by 38% from 13 dB with 10 wt% NiCF to 18 dB with 30 wt% NiCF. By side-feeding of NiCF, the EMI SE was enhanced about two times with 30 wt% NiCF compared to main-feeding of NiCF.

The results indicate that the EMI SE of SF-NiCF/PA6 composite was higher than that of its MF-NiCF/PA6 counterpart. The reason for this is that the MF-NiCF/PA6 composite can be subject to fiber length degradation and fiber damages, with it being exposed to the increased shearing force between the barrel screws during the extrusion process, compared to the SF-NiCF/PA6 composite. As a result, SF-NiCF/PA6 exhibited a fiber aspect ratio that was higher than the MF-NiCF/PA6 composite. The EMI SE of a carbon-fiber-reinforced composite material can be affected by the electrical pathway, which can be formed by the uniform distribution and three-dimensional network of carbon fibers in the composite. The long NiCF can provide a long pathway to connect the conductive carbon fibers to each other, making the NiCF/PA6 composite more effective for electromagnetic interference shielding.

## 4. Conclusions

The thermal stability, heat deflection temperature, storage modulus, tensile, flexural, surface resistivity, and EMI SE properties of NiCF/PA6 composites were significantly influenced by the length distribution of NiCF in the composites, depending on the feeding method (main-feeding or side-feeding) of chopped NiCF upon twin-screw extrusion processing.

NiCF/PA6 composites produced via side-feeding of NiCF exhibited thermal stability, HDT, and dynamic mechanical properties higher than those produced via main-feeding of NiCF depending on the NiCF content, with this being ascribed to the increased fiber length obtained by side-feeding of NiCF. The tensile and flexural properties were remarkably higher with the SF-NiCF/PA6 composites than with the MF-NiCF/PA6 counterparts, with them gradually increasing by increasing the NiCF up to 30 wt%. The surface resistivity and the EMI SE results were consistent with each other, depending on the NiCF content and the fiber-feeding method. Upon extrusion, side-feeding of NiCF significantly decreased the surface resistivity of the NiCF/PA6 composite and increased the EMI SE as well.

The present study stresses that one may highly enhance the composite properties with less fiber loading by side-feeding than by main-feeding of metal-coated carbon fiber during the twin-screw extrusion process.

**Author Contributions:** Conceptualization, writing—original draft preparation, writing—review and editing, supervision, and formal analysis, D.C.; methodology, investigation, and data curation, N.J. All authors have read and agreed to the published version of the manuscript.

**Funding:** This research received no external funding.

**Institutional Review Board Statement:** Not applicable.

**Informed Consent Statement:** Not applicable.

**Data Availability Statement:** The data presented in this study are available on request from the corresponding author.

**Conflicts of Interest:** The authors declare no conflict of interest.

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
