# Peer review of "Effect of Fiber Side-Feeding on Various Properties of Nickel-Coated Carbon-Fiber-Reinforced Polyamide 6 Composites Prepared by a Twin-Screw Extrusion Process"

_jcs, doi:10.3390/jcs7020068_

Round 1
Reviewer 1 Report
Very well written Manuscript, performed experiment very thoroughly described, and all data and assumptions for the experiment are given. The whole work ends with well-written conclusions.
My comments:
1. Figure 1 could possibly be enlarged to make the details stand out.
2. An electron microscopy image must be available in order to demonstrate the results of Figure 3 (Ηistogram).
A question that can be asked to the editors is: Do the imperfections of the composite material affect their results, and if so in what way?
The manuscript contains important scientific results.
Reviewer 2 Report
1. Please provide more details about current achievements in metal-coated fiber coextrusion in the introduction.
2. The texts in almost all figures, especially Fig. 1 and 2, were too small to read.
3. What's the structure of the die for the nickel-coated fiber coextrusion? Any details about the extrusion line? Cooling tank? vacuum chamber? Puller, etc.?
4. How to make the specimens for mechanical tests? Casting? If yes, how to cast?
5. Why the aspect ratio of NiCF was much higher? Any explanation?
6. Many in-depth discussions were needed. For example, why the length distribution and aspect ratio significantly affected the mechanical properties? The authors were expected to provide their own understanding, not only cited several paper to demonstrate this phenomenon was also observed by other researchers.
7. Were the properties, mechanical, electrical, etc., of the fibers fabricated in this work better than those of current fibers? Any comparison?
Reviewer 3 Report
jcs-2143013_ Review
Title: Effect of Fiber Side-Feeding on Various Properties of Nickel Coated Carbon Fiber-Reinforced Polyamide 6 Composites Prepared by Twin-Screw Extrusion Process
In peer-reviewed work the mechanical, thermal, electrical, and electromagnetic properties of nickel-coated carbon fiber-reinforced polyamide composites produced through twin-screw extrusion process were investigated. The manuscript corresponds to the Journal of Composites Science. The Introduction (Lines 25-85) and the list of references (Lines 546-621) are quite complete. The methodology of the study is described in sufficient detail.
The text is reasonably clear and easy to read.
All structural units of the manuscript are logically interconnected.
The manuscript contains important scientific results for practice, which can potentially serve as an incentive for further research into the technology of obtaining and properties of composite materials. Therefore, the manuscript is of interest to many specialists in this and related fields.
Comments and suggestions:
1. Lines 9-22. The purpose of the study, methods and results should be formulated more clearly.
2. Line 207-213. “Three-point flexural tests were also performed with neat PA6 and NiCF/PA6 composites with various fiber contents prepared by different fiber feeding methods using a universal testing machine (UTM, AG-50kNX, Shimadzu JP Co., Kyoto, Japan) according to the ASTM D790M standard [33]. The span-to-depth ratio was 32. The crosshead speed was 5.1 mm/min. The rectangular-shaped specimens had the dimensions of 12.5 mm × 12.5 mm × 3 mm. The average flexural modulus and strength were obtained from 10 specimens per each composite.”
It is excellent testing machine and good experiment. Since the article is of a scientific nature, a plot load-displacement would be interesting.
3. Line 324, Figure 5. NiCF contents = 0? A little technical editing is needed.
4. The main function of the studied composite is, as follows from the manuscript (lines 25-38), shielding of electromagnetic radiation. It is recommended to supplement the conclusions about the work done with quantitative estimates of the improvement of shielding, if the results of the presented study are used (line 324, Figure 10).
Reviewer
Round 2
Reviewer 1 Report
The suggested changes have been implemented.
Reviewer 2 Report
The authors answered all my questions in the revised manuscript. I have no comments.
Reviewer 3 Report
In peer-reviewed work the mechanical, thermal, electrical, and electromagnetic properties say of nickel-coated carbon fiber-reinforced polyamide composites produced through twin-screw extrusion process were investigated.
The manuscript contains important scientific results for practice, which can potentially serve as an incentive for further research into the technology of obtaining and properties of composite materials. Therefore, the manuscript is of interest to many specialists in this and related fields. By revising the manuscript, the paper quality was highly improved.